# Ballistic superconductivity in semiconductor nanowires

Hao Zhang[1,2,*], Önder Gül[1,2,*], Sonia Conesa-Boj[1,2,3], Michał P. Nowak[1,2,4], Michael Wimmer[1,2], Kun Zuo[1,2], Vincent Mourik[1,2], Folkert K. de Vries[1,2], Jasper van Veen[1,2], Michiel W.A. de Moor[1,2], Jouri D.S. Bommer[1,2], David J. van Woerkom[1,2], Diana Car[3], Sébastien R. Plissard[2,3], Erik P.A.M. Bakkers[1,2,3], Marina Quintero-Pérez[1,5], Maja C. Cassidy[1,2], Sebastian Koelling[3], Srijit Goswami[1,2], Kenji Watanabe[6], Takashi Taniguchi[6] & Leo P. Kouwenhoven[1,2,7]

Semiconductor nanowires have opened new research avenues in quantum transport owing to their confined geometry and electrostatic tunability. They have offered an exceptional testbed for superconductivity, leading to the realization of hybrid systems combining the macroscopic quantum properties of superconductors with the possibility to control charges down to a single electron. These advances brought semiconductor nanowires to the forefront of efforts to realize topological superconductivity and Majorana modes. A prime challenge to benefit from the topological properties of Majoranas is to reduce the disorder in hybrid nanowire devices. Here we show ballistic superconductivity in InSb semiconductor nanowires. Our structural and chemical analyses demonstrate a high-quality interface between the nanowire and a NbTiN superconductor that enables ballistic transport. This is manifested by a quantized conductance for normal carriers, a strongly enhanced conductance for Andreev-reflecting carriers, and an induced hard gap with a significantly reduced density of states. These results pave the way for disorder-free Majorana devices.

[1] QuTech, Delft University of Technology, 2600 GA Delft, The Netherlands. [2] Kavli Institute of Nanoscience, Delft University of Technology, 2600 GA Delft, The Netherlands. [3] Department of Applied Physics, Eindhoven University of Technology, 5600 MB Eindhoven, The Netherlands. [4] Faculty of Physics and Applied Computer Science, AGH University of Science and Technology, al. A. Mickiewicza 30, 30-059 Kraków, Poland. [5] Netherlands Organisation for Applied Scientific Research (TNO), 2600 AD Delft, The Netherlands. [6] Advanced Materials Laboratory, National Institute for Materials Science, 1-1 Namiki, Tsukuba 305-0044, Japan. [7] Microsoft Station Q Delft, 2600 GA Delft, The Netherlands. * These authors contributed equally to this work. Correspondence and requests for materials should be addressed to H.Z. (email: H.Zhang-3@tudelft.nl) or to Ö.G. (email: Gul.Onder@gmail.com) or to L.P.K. (email: L.P.Kouwenhoven@tudelft.nl).

Majorana modes are zero-energy quasiparticles emerging at the boundary of a topological superconductor[1–3]. Following proposals for their detection in a semiconductor nanowire coupled to a superconductor[4,5], several electron transport experiments reported characteristic Majorana signatures[6–14]. The prime challenge to strengthen these signatures and unravel the predicted topological properties of Majoranas is to reduce the remaining disorder in this hybrid system. Disorder can mimic zero-energy signatures of Majoranas[15–19], and results in states within the induced superconducting energy gap[20], the so-called soft gap, which renders the topological properties experimentally inaccessible[21,22]. The soft gap problem is attributed to the inhomogeneity of the hybrid interface[20,23–25] and has been overcome by a recent demonstration of epitaxial growth of Al superconductor on InAs nanowires[23], yielding a hard gap—a strongly reduced density of states within the induced superconducting gap. However, the Al-InAs nanowire system still contains residual disorder showing up in transport as unintentional quantum dots[13,23], a common observation in many previous instances of hybrid nanowire devices[9,18,19]. As an alternative material system, we have further developed the combination of InSb nanowires with NbTiN as our preferred choice of superconductor[6]. InSb is in general cleaner (that is, higher electron mobility[26–29]) than InAs. Moreover, InSb has a ~5 times larger g-factor, bringing down the required external magnetic field needed to induce the topological phase transition. Our preference for NbTiN relies on its high critical magnetic field exceeding 10 T.

Here we show ballistic superconductivity in InSb semiconductor nanowires. Our structural and chemical analyses demonstrate a high-quality interface between the InSb nanowire and a NbTiN superconductor. The high-quality interface enables ballistic transport manifested by a quantized conductance for normal carriers, and a strongly enhanced conductance for Andreev-reflecting carriers at energies below the superconducting gap. Our numerical analysis indicates a mean free path of several micrometres, implying ballistic transport of Andreev pairs in the proximitized nanowire. Finally, tunnelling conductance reveals an induced hard gap with a significantly reduced density of states. These results constitute a substantial improvement in induced superconductivity in semiconductor nanowires, and pave the way for disorder-free Majorana devices.

## Results

**Hybrid nanowire devices and their structural analysis.** We report on five devices with different geometries all showing consistent results. An overview of all the devices is given in Supplementary Fig. 1. Figure 1a,b shows a nanowire device consisting of a normal contact (Au), a nanowire (InSb) and a superconducting contact (NbTiN). This device was first measured at low temperature showing high-quality electron transport (data discussed below). After, the device was sliced open (using focused ion beam) and inspected sideways in a transmission electron microscope (TEM). The hexagonal facet structure of the nanowire is clearly visible (Fig. 1c and Supplementary Fig. 2). Except for the bottom facet that rests on the substrate, the polycrystalline superconductor covers the nanowire all around without any visible voids.

The precise procedure for contact realization is extremely important (see ref. 25). First, the native oxide at the InSb surface is wet-etched using a sulfur-based solution followed by an argon

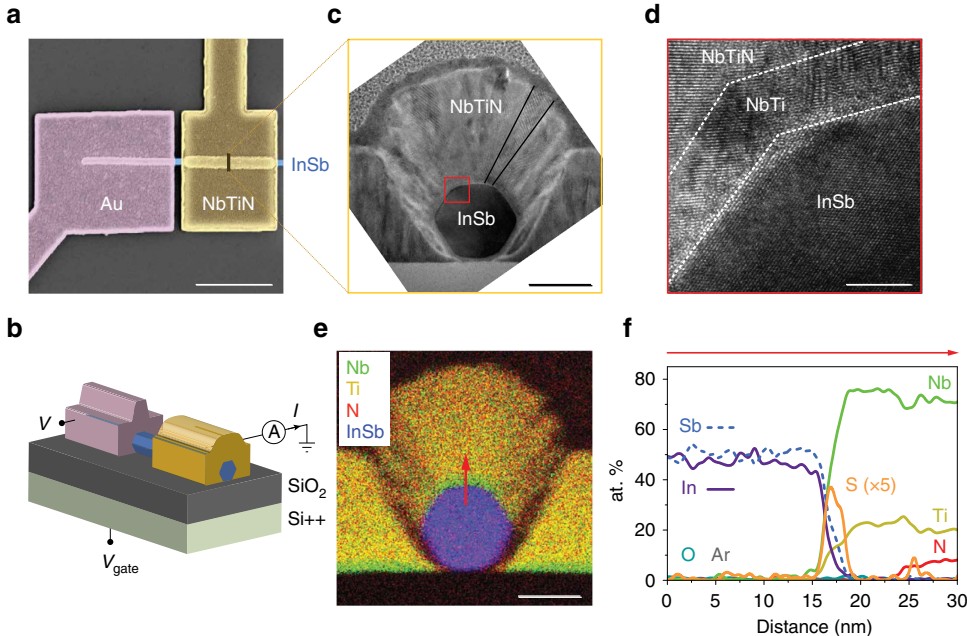

**Figure 1 | TEM analysis of a typical device.** (**a**) Top-view, false-colour electron micrograph of device A. Scale bar, 1 μm. Normal metal contact is Cr/Au (10 nm/125 nm) and superconducting contact is NbTi/NbTiN (5 nm/85 nm). Contact spacing is ~100 nm. (**b**) Device schematic and measurement setup. (**c**) Low-magnification high-resolution TEM (HRTEM) cross-sectional image from the device (see Methods). Scale bar, 50 nm. The cut was performed perpendicular to the nanowire axis, indicated by the dark bar in **a**. InSb nanowire exhibits a hexagonal cross-section surrounded by {220} planes. The NbTiN on the pre-layer NbTi crystallizes as cone-like elongated grains, indicated by the thin black lines. Corresponding fast Fourier transform confirms the polycrystalline character of the NbTiN region (Supplementary Fig. 2b). (**d**) HRTEM image near the interface (red square in **c**) shows that our cleaning procedure only minimally etches the wire and the InSb crystalline properties are preserved after the deposition. Scale bar, 5 nm. (**e**) Energy-dispersive X-ray (EDX) compositional map of the device cross-section. Scale bar, 50 nm. (**f**) EDX line scan taken across the interface as indicated by the red arrow in **e**. The sulfur content is multiplied by 5 for clarity. The system is oxygen and argon free (contact deposition is performed in an Ar plasma environment).

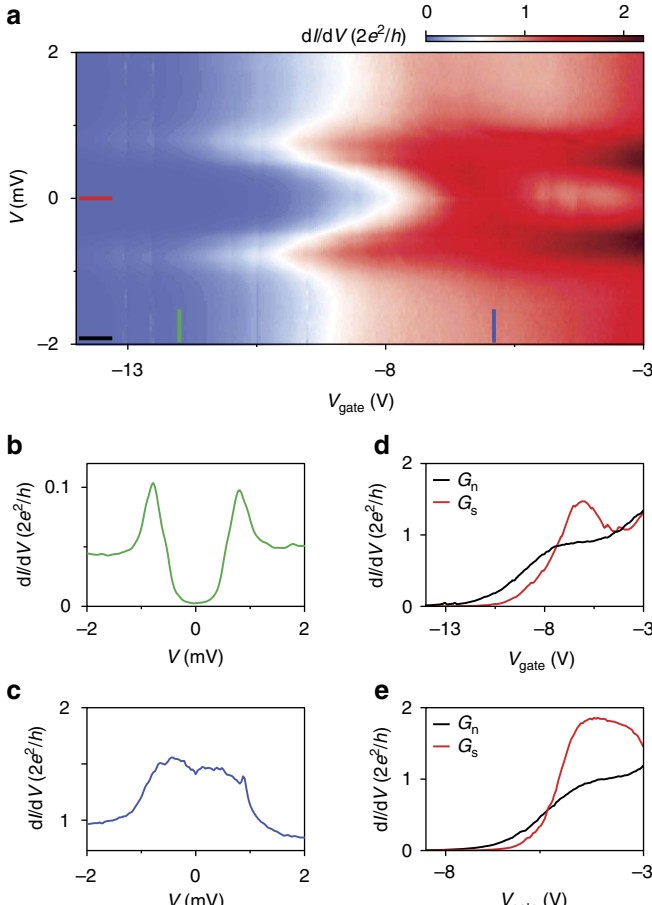

**Figure 2 | Ballistic transport at zero magnetic field.** (**a**) Differential conductance, d$I$/d$V$, as a function of bias voltage, $V$, and gate voltage, $V_{gate}$ for device B. (**b**) Vertical line cut from **a** in tunnelling regime (green trace, gate voltage = $-12$ V). (**c**) Vertical line cut from **a** on the conductance plateau (blue trace, gate voltage = $-5.9$ V). (**d**) Horizontal line cuts from **a** showing above-gap ($G_n$, black, $|V| = 2$ mV) and subgap ($G_s$, red, $V = 0$ mV) conductance. (**e**) Above-gap (black) and subgap (red) conductance for device C, where $G_s$ enhancement reaches $1.9 \times 2e^2/h$.

etch of sufficiently low power to avoid damaging the InSb surface (see Methods). The inclusion of sulfur at the interface results in band bending with electron accumulation near the surface of InSb[30] (Supplementary Fig. 3). Superconducting film deposition starts with NbTi, a reactive metal whose inclusion as a wetting layer is crucial to create a good electrical contact. Figure 1d shows that our cleaning procedure only minimally etches the wire and the InSb crystalline properties are preserved after the deposition (details in Supplementary Fig. 2). We detect a thin segregation layer ($\sim 2$ nm) between the polycrystalline NbTi and single-crystalline InSb. The chemical analysis (Fig. 1e,f) shows a material composition in agreement with our deposition procedure. More importantly, the inclusion of sulfur is clearly visible at the interface whereas the original native oxide is completely absent.

**Ballistic transport.** The high-quality structural properties in Fig. 1 result in largely improved electronic properties over the previous instances of hybrid nanowire devices. Figure 2a shows the differential conductance d$I$/d$V$ while varying the bias voltage $V$ between the normal and superconducting contacts, and

stepping the gate voltage $V_{gate}$ applied to the global back gate (Fig. 1b). We first of all note that throughout the entire gate voltage range in Fig. 2 we do not observe signs of the formation of unintentional quantum dots or any other localization effects resulting from potential fluctuations. Instead, we observe conductance plateaus at $2e^2/h$ for all devices, typical for ballistic transport and a clear signature of disorder-free devices. For a sufficiently negative gate voltage the non-covered nanowire section between normal and superconducting contacts is depleted and serves as a tunnel barrier. A vertical line cut from this regime is plotted in Fig. 2b, showing a trace typical for an induced superconducting gap with a strong conductance suppression for small $V$. The extracted gap value is $\Delta^* = 0.8$ meV. Increasing $V_{gate}$ first lowers and then removes the tunnel barrier completely. A vertical line cut from this open regime is plotted in Fig. 2c. In this case, the conductance for small $V$ is enhanced compared to the value above $\sim 1$ mV. Note that the range in $V$ showing an enhanced conductance in Fig. 2c corresponds to the same range showing the induced gap in Fig. 2b. The enhancement results from Andreev processes where an incoming electron reflects as a hole at the normal conductor-superconductor interface generating a Cooper pair[23,24,31,32]. This Andreev process effectively doubles the charge being transported from $e$ to $2e$ enhancing the subgap conductance. In Fig. 2c, the observed enhancement is by a factor $\sim 1.5$.

The Andreev enhancement is also visible in horizontal line cuts as shown in Fig. 2d. The above-gap conductance (black trace) taken for $|V| = 2$ mV represents the conductance for normal carriers, $G_n$. The subgap conductance, $G_s$, near $V = 0$ (Fig. 2d, red trace) shows an Andreev enhancement in the plateau region. Figure 2e shows a similar trace from another device where the enhancement in $G_s$ reaches $1.9 \times 2e^2/h$, very close to the theoretical limit: an enhancement factor of 2 in the case of a perfect interface. Finally, we note the dip in subgap conductance $G_s$ following the Andreev enhancement, observed both in Fig. 2d and Fig. 2e. The combined enhancement and dip structure provides a handle for estimating the remaining disorder by a comparison to theory, as discussed below.

**Theoretical simulation.** We construct a tight binding model of our devices (Fig. 3a) and numerically calculate the conductance using the Kwant package[33] (see Methods for details). In Fig. 3b, we plot the conductance traces obtained from the simulation for different disorder strength corresponding to varying mean free paths $l_e$. The calculated subgap conductance reproduces the dip structure observed in the experiment. We find that the dip is caused by mixing between the first and the second subband due to residual disorder (Supplementary Fig. 4). Even for weak disorder, subband mixing is strongly enhanced near the opening of the next channel, due to the van Hove singularity at the subband bottom. Hence, the Andreev conductance will generically exhibit a dip close to the next conductance step, instead of a perfect doubling. Figure 3c shows the measured subgap conductance $G_s$ and above-gap conductance $G_n$ for a device with a particularly flat plateau. Comparing Fig. 3b and Fig. 3c, we find good agreement for a mean free path of several micrometres. This implies ballistic transport of Andreev pairs in the proximitized wire section underneath the superconductor, whose length far exceeds the length of the non-covered wire between the contacts (see also Supplementary Fig. 5). Andreev enhancement allows for extracting mean free paths greatly exceeding the non-covered wire section since the subgap conductance is sensitive to even minute disorder in the proximitized wire section—a new finding of our study. This sensitivity is due to the quadratic dependence of the subgap

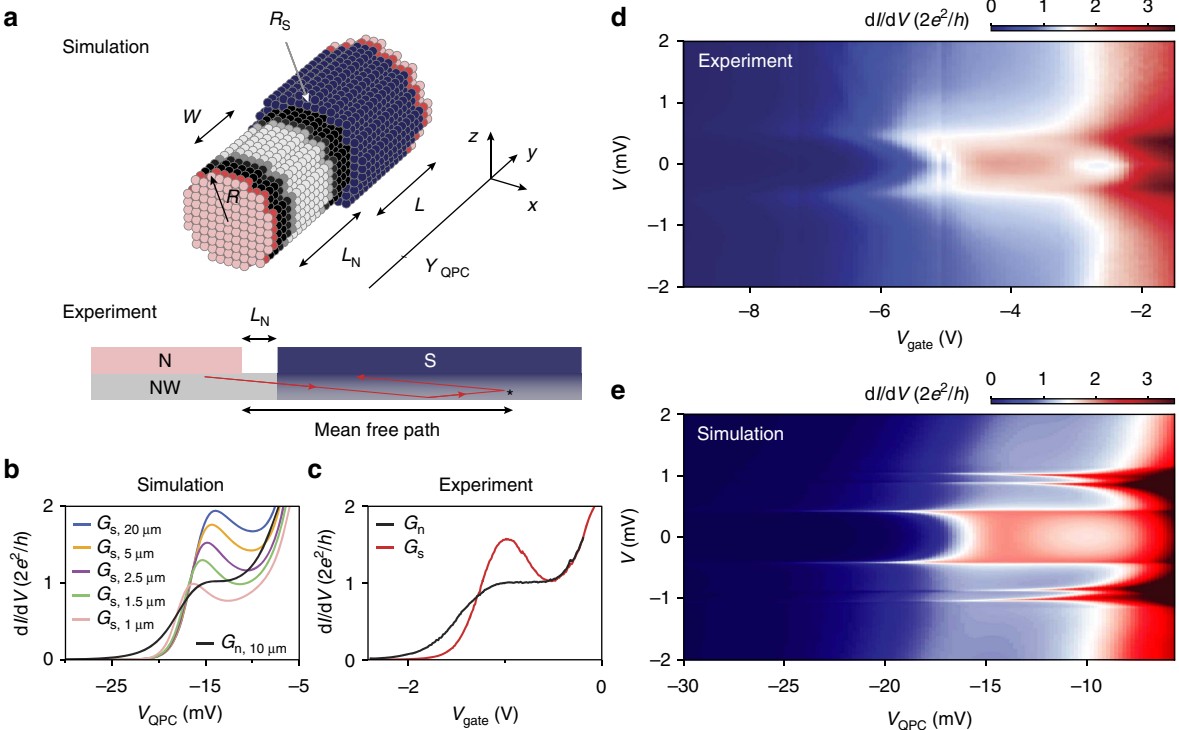

**Figure 3 | Theoretical simulation.** (**a**) Theoretical model (top): a cylindrical nanowire (black, grey, white) with length $L_N + L$ (100 nm + 800 nm), where the latter part is partially coated by a superconductor leaving the bottom surface uncovered. (Scheme shows $L = 100$ nm for clarity.) The wire radius $R$ is 40 nm and the superconducting film has a thickness $R_s = 10$ nm. (Our wire radius varies from device to device between 30 and 50 nm, and we have confirmed that our simulations give similar results within this range.) The wire is terminated from both sides with infinite leads (pink). Front lead is normal, back lead is normal/superconductor. Each little circle represents a three-dimensional mesh site with a size of 7 nm. White circles depict a potential barrier with a width $W = 60$ nm in the uncovered wire section forming a quantum point contact (QPC). Grey circles represent the smoothness of the barrier which is set to 5 nm. Experimental geometry (bottom): cross-sectional schematic shows the nanowire (NW), the normal contact (N) and the superconducting contact (S). Superconductivity is induced in the nanowire section underneath the superconducting contact. Transport is ballistic through a proximitized wire section, whose length far exceeds $L_N$, the length of the non-covered wire between the contacts. (**b**) Numerical simulation for devices with different mean free paths (see Supplementary Fig. 5). Black trace is for $G_n$ corresponding to a mean free path 10 μm, the rest are for $G_s$ corresponding to a mean free path ranging from 1 μm (pink) to 20 μm (blue). (**c**) Above-gap (black) and subgap (red) conductance for device D. (**d,e**) Comparison between the measurement (device C) and the simulation of a ballistic device with $l_e = 10$ μm. The induced superconducting gap edges for higher subbands, visible in the simulation as four symmetric peaks outside the gap around $V \sim \pm 1$ mV, are not observed in the experiment (see Methods for details).

conductance on the transmission probability (introduced below). In Fig. 3d,e, we compare a conductance measurement similar to the one in Fig. 2a with the simulation of a ballistic device. The overall agreement indicates a very low disorder strength for our devices.

**Hard superconducting gap.** The theory for electronic transport from a normal conductor via a quantum point contact to a superconductor was developed by Beenakker[31]. The subgap conductance is described by Andreev reflections[32], and for a single subband given by $G_s = 4e^2/h \times T^2/(2-T)^2$. The gate voltage-dependent transmission probability $T$ can be extracted from the measured above-gap conductance, given by $G_n = 2e^2/h \times T$. Figure 4a shows excellent agreement between the calculated and measured subgap conductance up to the point where the measured Andreev enhancement is reduced due to subband mixing. The highest transmission probability obtained from Andreev enhancement sets a lower bound on the interface transparency. Our typical enhancement factor of 1.5 (Figs 2d and 3c) implies an interface transparency $\sim 0.93$ and our record value of 1.9 (Fig. 2e) gives a transparency larger than 0.98 (see Measurement setup and data analysis in Methods).

The comparison between $G_s$ versus $G_n$ can be continued into the regime of an increasing tunnel barrier. Figure 4b,c show traces of $dI/dV$ for successively lower conductances. The subgap conductance suppression reaches $G_s/G_n \sim 1/50$, a value comparable to the results obtained with epitaxial Al[23]. A comparison between the measured subgap conductance and Beenakker's theory (without any fit parameters) is shown in Fig. 4d. The excellent agreement over three orders of magnitude in conductance implies that the subgap conductance is very well described by Andreev processes and no other transport mechanisms are involved[23,24]. The lowest conductance ($\sim 5 \times 10^{-4} \times 2e^2/h$) reaches our measurement limit, causing the deviation from theory. The inset to Fig. 4b shows how the subgap conductance increases when applying a magnetic field. Finally, in Supplementary Fig. 6 we show the magnetic field dependence of the induced gap and Andreev enhancement for a magnetic field along the nanowire axis. We again find a subgap conductance increasing with magnetic field, and an Andreev enhancement vanishing at a magnetic field ($< 1$ T) smaller than the critical field of our NbTiN film. We speculate that the increasing subgap conductance and the decreasing Andreev enhancement are due to vortex formation in our NbTiN film, a type-II superconductor. Future studies should be directed

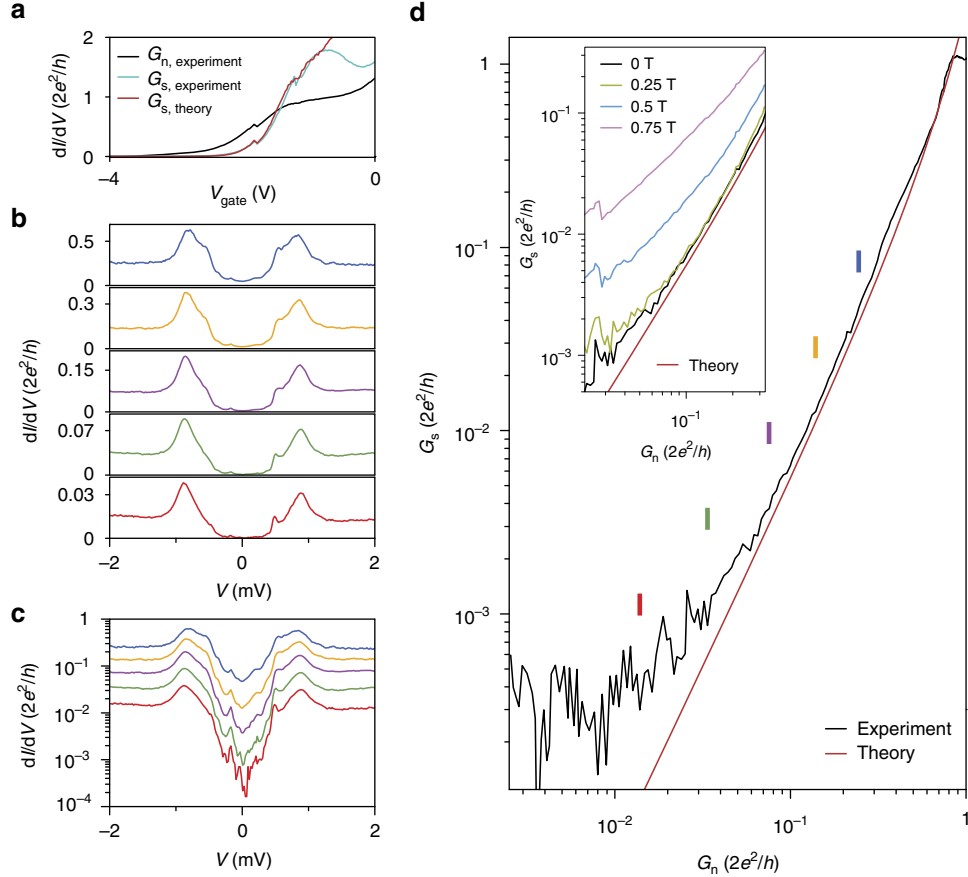

**Figure 4 | Hard gap and Andreev transport.** (**a**) Above-gap (black) and subgap (blue) conductance for device E. Red curve is a theory prediction based on single channel Andreev reflection, agreeing perfectly with experimental data without any fitting parameter up to the dip on the right side of the plateau where the second channel starts conducting. (**b,c**) Five typical gap traces corresponding to the five colour bars indicated in **d** plotted on a linear and logarithmic scale. The subgap conductance is suppressed by a factor up to 50 for the lowest conductance (red trace). (**d**) Subgap conductance $G_s$ as a function of above-gap conductance $G_n$ for device A. Red curve is the theory prediction assuming only Andreev processes. Inset shows $G_s$ versus $G_n$ taken at different magnetic fields.

towards developing a quantitative description of such magnetic field-induced deviation from Andreev transport, whose understanding plays a crucial role in realizing a topological quantum bit based on semiconductor nanowires.

## Methods

**Nanowire growth and device fabrication.** InSb nanowires have been grown by Au-catalysed vapour–liquid–solid mechanism in a metal organic vapour phase epitaxy reactor. The InSb nanowire crystal direction is [111] zinc blende, free of stacking faults and dislocations[34]. Nanowires are deposited one-by-one using a micro-manipulator[35] on a substrate covered with 285 nm thick $SiO_2$ serving as a gate dielectric for back-gated devices. For local-gated device D, extra set of bottom gates are patterned on the substrate followed by transfer of h-BN ($\sim$30 nm thick) onto which nanowires are deposited. The contact deposition process starts with resist development followed by oxygen plasma cleaning. Then, the chip is immersed in a sulfur-rich ammonium sulfide solution diluted by water (with a ratio of 1:200) at 60 °C for half an hour[36]. At all stages care is taken to expose the solution to air as little as possible. For normal metal contacts[27], the chip is placed into an evaporator. A 30 s Helium ion milling is performed *in situ* before evaporation of Cr/Au (10 nm/125 nm) at a base pressure $<10^{-7}$ mbar. For superconducting contacts[25], the chip is mounted in a sputtering system. After 5 s of *in situ* Ar plasma etching at a power of 25 W and an Ar pressure of 10 mTorr, 5 nm NbTi is sputtered followed by 85 nm NbTiN.

**Measurement setup and data analysis.** All the data in this article is measured in a dilution refrigerator with a base temperature of around 50 mK using several stages of filtering. The determination of the Andreev enhancement factor depends sensitively on the contact resistance subtracted from the measured data. In all our analysis, we only subtract a fixed-value series resistance of 0.5 kΩ solely to account

for the contact resistance of the normal metal lead. This value is smaller than the lowest contact resistance we have ever obtained for InSb nanowire devices[27], which makes the values for the interface transparency a lower bound.

**Structure characterization.** The cross-section and lamella for TEM investigations were prepared by focused ion beam (FIB). FIB milling was carried out with a FEI Nova Nanolab 600i Dualbeam with a Ga ion beam following the standard procedure[37]. We used electron induced Co and Pt deposition for protecting the region of interest and a final milling step at 5 kV to limit damage to the lamella. High-resolution TEM (HRTEM) and scanning TEM analyses were conducted using a JEM ARM200F aberration-corrected TEM operated at 200 kV. For the chemical analysis, energy-dispersive X-ray measurements were carried out using the same microscope equipped with a 100 mm² energy-dispersive X-ray silicon drift detector (SSD).

**Characterization of NbTiN.** Our NbTiN films are deposited using an ultrahigh vacuum AJA International ATC 1800 sputtering system (base pressure $\sim 10^{-9}$ Torr). We used a $Nb_{0.7}Ti_{0.3}$ wt.% target with a diameter of 3 inches. Reactive sputtering resulting in nitridized NbTiN films was performed in an Ar/N₂ process gas with 8.3 at.% N₂ content at a pressure of 2.5 mTorr using a DC magnetron sputter source at a power of 250 W. An independent characterization of the NbTiN films gave a critical temperature of 13.3 K for 90 nm thick films with a resistivity of 126 μΩ·cm and a compressive stress on Si substrate.

**Details of the theoretical simulation.** The system is described by the spin-diagonal Bogoliubov–de Gennes Hamiltonian

$$H = \left(\frac{\hbar^2 \mathbf{k}^2}{2m^*} - \mu + V(x,y,z)\right)\tau_z + \Delta(x,y,z)\tau_x, \qquad (1)$$

acting on the spinor $\Psi = (\psi_{e+}, \psi_{e-}, \psi_{h-}, -\psi_{h+})^T$. The Pauli matrices act on the electron-hole degree of freedom. Potential in the nanowire is described by $V(x, y, z) = \tilde{V}_{qpc}(y) + V_D(x, y, z)$, where $\tilde{V}_{qpc}(y)$ describes a quantum point contact given by

$$\tilde{V}_{qpc}(y) = -\frac{eV_{QPC}}{2}\left[\tanh\frac{y - Y_{QPC} + W/2}{\lambda} - \tanh\frac{y - Y_{QPC} - W/2}{\lambda}\right].$$

Here $Y_{QPC}$ is the centre position of the barrier (Fig. 3a). Barrier width is $W = 60$ nm, and the barrier height is controlled by $V_{QPC}$. The softness of the barrier is given by $\lambda$ which we take 5 nm. $V_D(x, y, z)$ accounts for disorder, which is modelled as a spatially varying potential with random values from a uniform distribution within a range $[-U_0, U_0]$ where amplitude $U_0 = \sqrt{3\pi/l_e m^{*2} a^3}$ is set by mean free path $l_e$.

We approximate the superconductor covering the wire by a layer of non-zero $\Delta$ for $(x^2 + z^2) > R$ and $y > L_N$ and $z > -R$. The huge wave vector difference in the superconductor and semiconductor cannot be captured in a numerical simulation of a three-dimensional device. Hence, to capture the short coherence length in the superconductor, we take a superconducting shell of thickness $R_S = 10$ nm and $\Delta = 200$ meV. We then tune the induced gap to be close to the experimental value ($\sim 0.5$ meV) by reducing the hopping between the semiconductor and the superconductor by a factor of 0.8.

The transport properties of the system are calculated using Kwant package[33] with the Hamiltonian in equation (1) discretized on a three-dimensional mesh with spacing $a = 7$ nm and infinite input (normal) and output (normal/superconducting) leads. For a given $V_{QPC}$ and excitation energy $\varepsilon$ we obtain the scattering matrix of the system from which we subsequently extract electron $r_e(\varepsilon)$ and hole $r_h(\varepsilon)$ reflection submatrices. Finally, we calculate thermally averaged conductance for injection energy $E = -eV$ according to

$$G(E) = \int d\varepsilon \mathcal{G}(\varepsilon)\left(-\frac{\partial f(E, \varepsilon)}{\partial \varepsilon}\right),$$

where the Fermi function

$$f(E, \varepsilon) = \frac{1}{e^{(\varepsilon - E)/k_b T} + 1},$$

and $\mathcal{G}(\varepsilon) = N - ||r_e(\varepsilon)||^2 + ||r_h(\varepsilon)||^2$. We assume chemical potential to be $\mu = 30$ meV, which gives $N = 3$ spin-degenerate modes in the leads. The presented results are obtained for $T = 70$ mK and InSb effective mass $m^* = 0.014 m_e$.

**Data availability.** All data are available at http://doi.org/10.4121/uuid:fdeb81 ab-1478-4682-9f48-dec1c83242bd (ref. 38). The code used for the simulations is available upon request.

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

## Acknowledgements

We thank A.R. Akhmerov, O.W.B. Benningshof, A. Geresdi, J. Kammhuber and A.J. Storm for discussions and assistance. This work has been supported by the Netherlands Organisation for Scientific Research (NWO), Foundation for Fundamental Research on Matter (FOM), European Research Council (ERC) and Microsoft Corporation Station Q.

## Author contributions

H.Z. and Ö.G. fabricated the devices, performed the measurements and analysed the data. S.C.-B. performed the TEM analysis. M.P.N. and M.W. performed the numerical simulations. K.Z., V.M., F.K.d.V., J.v.V., M.W.A.d.M., J.D.S.B., D.J.v.W., M.Q.-P., M.C.C. and S.G. contributed to the experiments. D.C., S.P. and E.P.A.M.B. grew the InSb nanowires. S.K. prepared the lamellae for the TEM analysis. K.W. and T.T. synthesized the h-BN crystals. L.P.K. supervised the project. All authors contributed to the writing of the manuscript.

## Additional information

**Competing interests:** The authors declare no competing financial interests.

