## [Peer Review File · Nature Communications]

REVIEWERS' COMMENTS:

Reviewer #2 (Remarks to the Author):

The main concern in the previous referee reports was the weak connection between the claims in the second part of the manuscript (zero bias peaks) and the demonstration of ballistic superconductivity in the first part. Now, this weakness has been corrected since the authors decided to remove all these results concerning zero bias peaks in this revised manuscript and just focus on ballistic superconductivity.

My overall impression of this new version of the manuscript is very good and the claim of ballistic superconductivity in proximitized InSb is, in my opinion, well justified: Apart from the demonstration of high-quality interfaces by means of thorough structural and chemical analyses, the authors demonstrate ballistic transport in the superconducting regime by means of a strong Andreev-enhanced conductance. Moreover, the hard gaps in the tunneling regime, with a greatly reduced subgap density of states, clearly show the good quality of the proximity effect. Given the relevance of these experimental improvements for Majorana detection in nanowires, I recommend publication.

REVIEWERS' COMMENTS:

Reviewer #2 (Remarks to the Author):

The main concern in the previous referee reports was the weak connection between the claims in the second part of the manuscript (zero bias peaks) and the demonstration of ballistic superconductivity in the first part. Now, this weakness has been corrected since the authors decided to remove all these results concerning zero bias peaks in this revised manuscript and just focus on ballistic superconductivity.

We agree with the reviewer in their statement above.

My overall impression of this new version of the manuscript is very good and the claim of ballistic superconductivity in proximitized InSb is, in my opinion, well justified: Apart from the demonstration of high-quality interfaces by means of thorough structural and chemical analyses, the authors demonstrate ballistic transport in the superconducting regime by means of a strong Andreev-enhanced conductance. Moreover, the hard gaps in the tunneling regime, with a greatly reduced subgap density of states, clearly show the good quality of the proximity effect. Given the relevance of these experimental improvements for Majorana detection in nanowires, I recommend publication.

We thank the reviewer for helping us improve the manuscript in the previous rounds of review.